

# Newly discovered series of meteorological measurements in SW Greenland (Nuuk) in the period 1806–13

Rajmund Przybylak[1,2], Andrzej Araźny[1,2], Przemysław Wyszyński[1,2], Garima Singh[1], and Konrad Chmist[1]

[1] Faculty of Earth Sciences and Spatial Management, Nicolaus Copernicus University, Toruń, Poland, rp11@umk.pl
[2] Centre for Climate Change Research, Nicolaus Copernicus University, Toruń, Poland, cccr@umk.pl

Corresponding authors: Garima Singh, garima.singh@doktorant.umk.pl
Przemysław Wyszyński, przemyslaw.wyszynski@umk.pl

**Abstract.** The article presents a description of a newly discovered, unique series of meteorological measurements in SW Greenland (Godthåb [now Nuuk]) from the beginning of the 19th century (1 November 1806 to 16 August

1813). The series is the longest available from before 1840, not only for Greenland but also for the entire Arctic. The handwritten meteorological register was discovered in the archives of the Royal Society in London (MA/154). The meteorological observations were carried out by the German mineralogist Dr Charles Lewis Giesecke. The observations include measurements, taken two to three times per day, of air temperature, atmospheric pressure and wind direction. In addition, the meteorological register briefly describes the weather conditions for each day. In

the article, we present a detailed analysis of thermal conditions for the period covered by a complete series of measurements (Aug 1807–Jul 1813). The analysis of air temperature clearly shows that the study period was one of the coldest periods (possibly the coldest) in the past two millennia. A cooling of this severity has previously been found for the study region, the whole of Greenland and the whole Arctic. Among the available reconstructions that use different proxy data or that use climate models for this purpose, most of the reconstructions of air

temperature are almost fully consistent with the available results of meteorological observations for this period. Intense volcanic activity and, to a lesser degree, the low solar activity connected with the Dalton minimum are most often given as reasons for the cooling of the early 19th century.

**Keywords**: Arctic, Greenland, historical climatology, data rescue, climate records, temperature

## 1. Introduction

The main aim of historical climatologists in recent decades has been to reconstruct the climate on different spatial scales – from global to regional and local – for the last centuries. For this purpose, documentary evidence is used from before the beginning of regular meteorological

observations (Pfister et al., 1999; Brázdil et al., 2005), which is usually taken as the year 1850, except for Africa and the Arctic (for which 1890 is the usual start point) (Brönnimann et al., 2019). One type of very useful information about past weather and climate is an early, irregular, isolated series of meteorological measurements. Recently, an initiative was created to inventory all available data of this type (including existing databases) for the whole world, including the

Arctic, and then make it available in digital form (Brönnimann et al., 2019; Lundstad et al., 2023). This activity has found recognition and interest among many scientists and has been called "data rescue activity". For the Arctic as a whole and for its different regions (Canadian Arctic, Greenland, Svalbard and Novaya Zemlya), a summary of existing meteorological measurements was conducted by Przybylak and Vizi (2005), Vinther et al. (2006), Przybylak



et al. (2010, 2016, 2018), and Przybylak and Wyszyński (2017). Queries conducted in many
      European and Canadian archives and libraries allowed the collection of 118 series of average
      monthly air temperature values from the years 1801–1920 (for details, see Table I in Przybylak
      et al. 2010), with the greatest amount dating from after the 1st International Polar Year 1882/83.
      The majority of the gathered series (77.1%) cover periods of less than two years; however,
series of one year or less dominate (58.5%) (Przybylak et al., 2010).

             Meteorological data from prior to 1850 are rare and mainly available for Europe and
      eastern North America (see Fig. 2a in Brönnimann et al. 2019). According to the map presented
      in this article (Fig. 2a), such a series for Greenland exists only for the south-western coast.
      Detailed descriptions of the observations available for this time are provided in papers by
Vinther et al. (2006) and, recently, for the late 18th century by Demarée et al. (2020), Demarée
      and Ogilvie (2021) and Przybylak et al. (2024). According to these publications, the oldest
      series of observations of about one year or more is that for Neu-Herrnhut (now Nuuk) for the
      expedition year Sep 1767–July 1768. All known meteorological observations made here in the
      late 18th century were usually performed by the Moravian missionaries (for more information
about their activity, see, e.g., Lüdecke (2005), Demarée and Ogilvie (2008), or Demarée et al.
      (2020). They are usually available as meteorological registers (some have survived and are
      available in archives in Germany, the UK and the USA; see Demarée and Ogilvie (2008) for
      more details), and some have also been published in annual reports (diaries handwritten in Old
      German or English). The latter sources, which contain only sparse and irregular measurement
data relating to only a few parameters, have been assessed in terms of their utility in application
      to climate studies (particularly for the study of weather extremes in SW Greenland) by Kodzik
      (2019) and Borm et al. (2021). Any new data series from the early instrumental period,
      especially for the Arctic (including Greenland), is very precious for calibrating and
      reconstructing climate using proxy data from so-called natural archives (ice cores, tree rings,
lacustrine sediments, etc.). Vinther et al. (2006) document this fact very clearly, showing a
      strong correlation (0.6–0.7) between reconstructed SW Greenland winter temperatures and ice-
      core winter-season proxy. Although they did enormous work rescuing a lot of early instrumental
      data for SW Greenland, they found neither the reportedly oldest meteorological data series for
      Nuuk relating to Sep 1767–July 1768 (see their Fig. 2, where the series starts at 1784) nor the
series newly discovered by us for 1806–13. This latter, entirely new series contains complete
      data from six years of observations and is the longest series available for the Arctic (including
      Greenland) for the period before 1840. For the 1840s and 1850s, we have a continuous series
      of data available for Illulisat (Jacobshavn) (Vinther et al., 2006) and probably also for Lichtenau



(1843–51) and Neu-Herrnhut (1843–60) according to Table 2 (Lüdecke et al., 2005). To check the completeness of the last two series, we need to have access to the so-called Lamont collections (kept at the Chair of Ecoclimatology at the Technical University of Munich). Despite numerous requests, the owner of the collection does not want to make it available to us.

From this brief review of the state of knowledge about early meteorological observations available for Greenland, it is clear that many exist. Despite many years of searches for such observations by many Danish and UK scientists (Vinther et al., 2006) and by Polish researchers and climatologists (Przybylak et al., 2010), no one has managed to obtain information about the existence of long-term continuous meteorological observations in Nuuk for the period between 1806 and 1813. While searching the London archives for early instrumental meteorological observations by the Moravian Brothers, we happened upon this series. Therefore, the main goal of this article is to present this newly discovered series of observations to a broader audience of scientists and to present the first results of climate analysis. This analysis is limited, however, to a description of conditions and changes in air temperature in Nuuk at that time. (Descriptions of other parameters will follow in future publications.) The secondary goal is to compare air temperatures in the study period against earlier temperature observations from the late 18th century (Przybylak et al., 2024) and with later records, including modern observations from 1991–2020.

## 2. Area, data and methods

The meteorological observations analyzed here were made at the beginning of the 19th century in the territory of present-day Nuuk (Danish name *Godthåb*, $\varphi$= 64° 11' 0.49" N, $\lambda$= -51° 43' 17.65" W; see also Fig. 1), the capital and largest city of Greenland. According to the title page of the manuscript and the Royal Society catalogue describing the source, the scientist responsible for making them was the German mineralogist Dr Charles Lewis Giesecke (born Johann Georg Metzler [1761–1833]) (Fig. 2). However, given the numerous biographical accounts of his stay in Greenland (e.g., Monaghan 1993; Jørgensen, 1996; Wyse Jackson, 1996; Whittaker, 2001), there is some doubt as to whether he conducted the meteorological observations alone; he may have had help from the local Inuit community and Moravian missionaries.





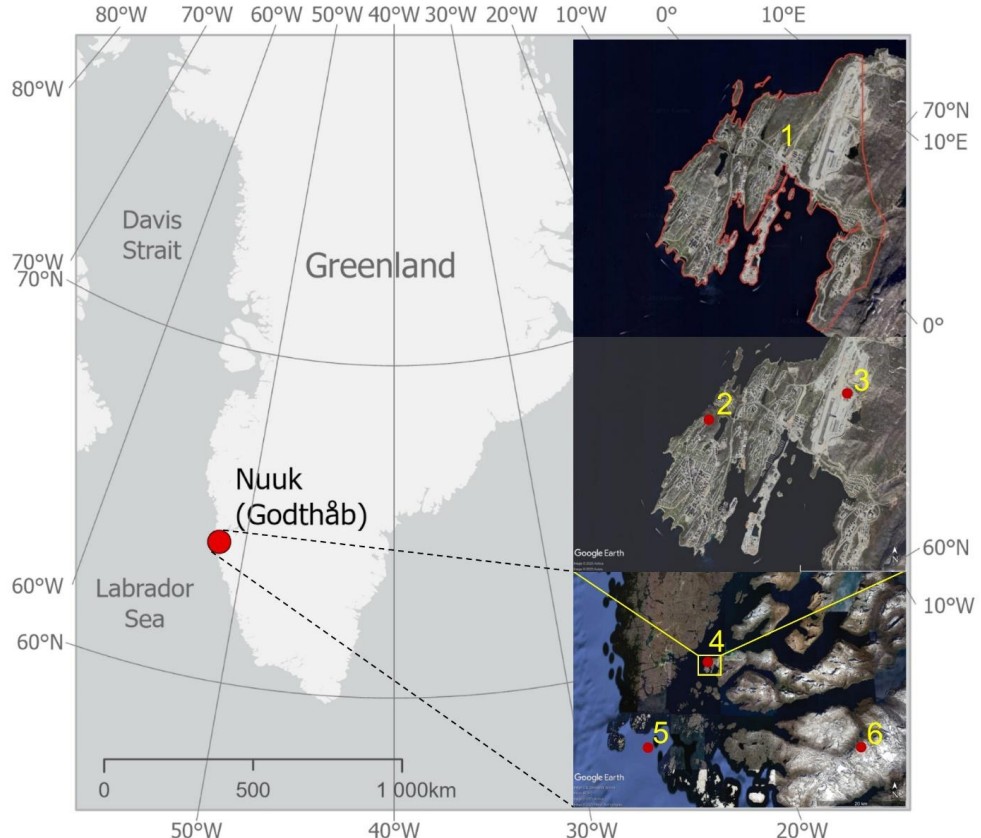

**Fig. 1.** Location of historical and contemporary sites of meteorological measurements in SW Greenland. Explanation: 1 – historical site Godthåb (1806–13, the exact location is unknown); 2 – 4250 Nuuk (1991–2020); 3 – 4254 Mittarfik Nuuk (2001–20), 4 – ModE-RA (1806–13), 5 – 20CRv3_C (1806–13, coastal grid point), 6 - 20CRv3_T (1806–13, terrestrial grid point). Map data for location of sites: © Google Earth; images © U.S. Geological Survey, © IBCAO, © 2025 Maxar Technologies, © 2025 Airbus and © 2025 Asiaq





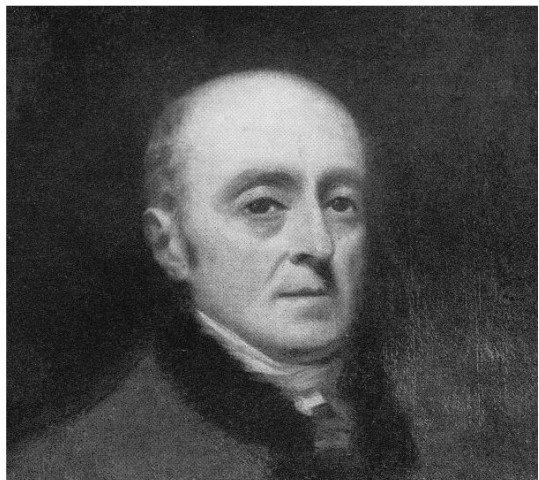

**Fig. 2.** Portrait of Sir Charles Lewis Giesecke (1761–1833) painted by the Scottish painter Raeburn in 1817 (after
Whittaker, 2001)


Dr Charles Lewis Giesecke arrived at Friederichshaab in Greenland on May 31, 1806
(Whittaker, 2001). The primary purpose of his research visit and stay in SW Greenland, which
was planned to last for one to two years, was to conduct a geological reconnaissance,
particularly in search of new minerals. We know that the meteorological observations he began
on November 1, 1806, continued regularly until April 1807. From then until August of the same
year, the observations were intermittent and irregular. Therefore, he likely carried out most of
his geological research from June to October 1806 and then from May to July 1807. There is
also evidence that he conducted studies of minerals in the summers of 1808 and 1809
(Whittaker, 2001), but these trips were probably shorter. During this time, he organised research
expeditions along the south-western coast of Greenland, from Upernavik in the north to Cape
Farewell in the south (Whittaker, 2001). To reach these places, he mostly used boats, but also
sledges and travelled on foot. Due to the onset of the Napoleonic War in 1807, much of Europe
was occupied by French troops. Thus, Giesecke could not return to Denmark and decided to
stay in Greenland until the political situation changed and allowed him to return. This fact likely
led to the initiation of systematic meteorological observations, which were conducted over six
years (Aug 1807 – Jul 1813), resulting in an almost complete series of observations made three
times a day (morning, midday and evening). The data used in this study were taken from the
manuscript MA/154 found in the Archives of the Royal Society in London (Fig. 3). As
mentioned above, it is possible that, for these measurements, especially during his exploratory



travels, he involved Inuits and (especially) the Moravian missionaries. These latter were already present in the area and had extensive experience in conducting meteorological observations (see e.g., Demarée and Ogilvie 2008; Przybylak et al., 2024).

As Fig. 3 shows, Giesecke conducted measurements and observations of the following
meteorological variables: atmospheric pressure (morning and evening), air temperature (morning, midday and evening) and wind direction (morning and evening). The precise times of observations are not given. In addition to the meteorological data available in the meteorological registers, he briefly described the weather conditions for each day. At the end of each month, he also included a short summary of the weather conditions (at the bottom of
the table of meteorological data). Unfortunately, the register does not provide information about the units used for the measurements or details about the thermometer's exposure. It is assumed that the thermometer was placed on the north-facing wall outside Giesecke's building. However, in the Arctic, where there is polar day during summer months, such placement does not eliminate the influence of solar radiation unless the thermometer is adequately shielded. The
detailed location of the place where observations were probably done (whether it was near the sea or at some distance from the sea, its elevation, etc.) is also unknown.

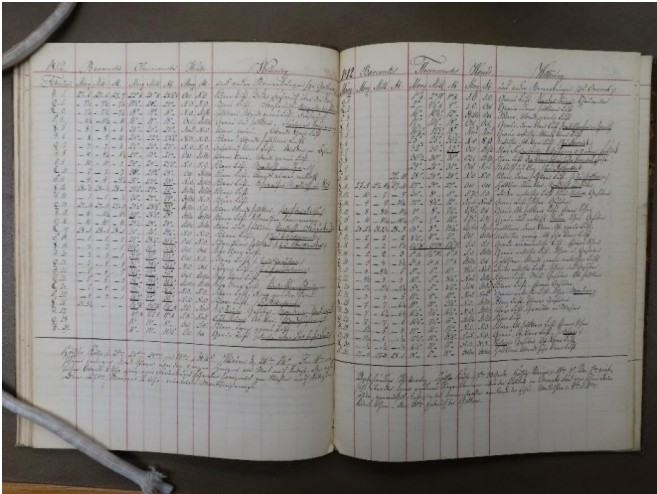

**Fig. 3.** Example of a manuscript presenting meteorological observations in Godthåb (1 Feb 1806–16 Aug 1813) (data presented in the manuscript: 1 February 1812 to 31 March 1812, source – Archives of the Royal Society in
London, MA/154)

In the article, we present an analysis limited to air temperature – the most important variable in every climate analysis. All source data were quality controlled by visual inspection, but no wrong or suspect data were found. As we wrote earlier, the unit in which air temperature measurements are presented in the meteorological register is not given. To determine which



unit was probably used in Giesecke's measurements, we compared his results against the
parallel temperature observations available for another location in Godthåb during 1811–12
(Vinther et al., 2006). Based on these, we concluded that temperatures were measured in degrees
Celsius. Although the exact times of air temperature measurements are unknown, we calculated
mean daily air temperature (MDAT) using a weighted average: (Tmorning + Tmidday + 2 ×

Tevening) / 4. To check the probable biases in the calculation of MDATs, we assessed them
using hourly air temperature data taken from the Nuuk 4250 station from 2010 to 2020 (Drost
Jensen, 2022). We calculated MDATs according to eight different formulas:

$$\text{MDAT1} = (T1 + T2 + T3, \ldots, T24)/24 \qquad (1)$$

$$\text{MDAT2} = (T6 + T12 + T18)/3 \qquad (2)$$

$$\text{MDAT3} = (T7 + T13 + T19)/3 \qquad (3)$$

$$\text{MDAT4} = (T8 + T14 + T20)/3 \qquad (4)$$

$$\text{MDAT5} = (T8 + T14 + 2*T20)/4 \qquad (5)$$

$$\text{MDAT6} = (T8 + T14 + 2*T21)/4 \qquad (6)$$

$$\text{MDAT7} = (T6 + T12 + 2*T20)/4 \qquad (7)$$

$$\text{MDAT8} = (T6 + T12 + 2*T21)/4 \qquad (8)$$

where MDAT1 is calculated from 24 hourly measurements per day (the "real daily average").

In the next step, we compared the MDAT1 results against those obtained using all other
formulas (MDAT2–MDAT8, which rely on only three measurement times per day) (see Table
1). Analysis shows that, in the cold half-year (Oct–Mar), the maximum bias reaches 0.1 °C,

whereas in summer the bias is highest but does not exceed 0.5 °C. The formulas MDAT6,
MDAT7 and MDAT8 are closer to MDAT1, with a bias usually of 0.0 °C, except for June, July
and August, when the bias maximally reaches 0.2 °C.

**Table 1**. Mean monthly and annual differences in air temperature (°C) between MDATs calculated according to
formulas MDAT2–MDAT8 in comparison to the average calculated using the MDAT1 formula

| Period | Jan | Feb | Mar | Apr | May | Jun | Jul | Aug | Sep | Oct | Nov | Dec | Year |
|--------|-----|-----|-----|-----|-----|-----|-----|-----|-----|-----|-----|-----|------|
| MDAT2-MDAT1 | 0.0 | 0.0 | 0.1 | 0.1 | 0.3 | 0.3 | 0.4 | 0.3 | 0.1 | 0.1 | 0.0 | 0.0 | 0.15 |
| MDAT3-MDAT1 | 0.0 | 0.1 | 0.0 | 0.1 | 0.3 | 0.4 | 0.5 | 0.3 | 0.1 | 0.1 | 0.1 | 0.0 | 0.17 |
| MDAT4-MDAT1 | 0.0 | 0.1 | 0.0 | 0.2 | 0.3 | 0.4 | 0.5 | 0.4 | 0.2 | 0.1 | 0.0 | 0.0 | 0.18 |
| MDAT5-MDAT1 | 0.0 | 0.1 | 0.0 | 0.1 | 0.3 | 0.4 | 0.5 | 0.3 | 0.2 | 0.1 | 0.0 | 0.0 | 0.17 |
| MDAT6-MDAT1 | 0.0 | 0.1 | 0.0 | 0.1 | 0.1 | 0.2 | 0.2 | 0.1 | 0.1 | 0.0 | 0.0 | 0.0 | 0.07 |
| MDAT7-MDAT1 | 0.0 | 0.0 | 0.0 | 0.0 | 0.1 | 0.2 | 0.2 | 0.0 | 0.0 | 0.0 | 0.0 | 0.0 | 0.06 |
| MDAT8-MDAT1 | 0.0 | 0.0 | 0.0 | 0.0 | 0.0 | 0.0 | -0.1 | -0.2 | -0.1 | 0.0 | 0.0 | 0.0 | -0.04 |
| **Max bias** | **0.0** | **0.1** | **0.1** | **0.2** | **0.3** | **0.4** | **0.5** | **0.4** | **0.2** | **0.1** | **0.1** | **0.0** | **0.18** |



None of the possible biases in air temperature measurements that are associated with the type and accuracy of the thermometer used or its exposure can be estimated due to the lack of relevant information. However, the exposure bias should be minimal or zero for the polar night due to the absence of solar radiation at such times.

The MDAT values, which are available at https://doi.org/10.18150/IGYNGV
(Przybylak et al., 2025), were used to calculate standard (monthly, seasonal and annual means) and less-typical climate statistics (indices) such as: 1) day-to-day temperature variability (DDTV); 2) frequency of occurrence of MDAT in 1-degree intervals, including the calculations of skewness ($\gamma1$) and kurtosis ($\gamma2$) of analyzed sets of air temperature using formulas recommended by von Storch and Zwiers (1999); 3) thermal seasons after Baranowski (1968)
proposition for polar regions (for details, see Przybylak et al. 2024); 4) thermal roses of wind (relation between wind direction and temperature); and 5) annual air temperature range (ATR) and thermal climate continentality (K) using the following formula proposed by Ewert (1972):

$$K = [ATR – (3.81·\sin\varphi) + 0.1)] / (38.39·\sin\varphi + 7.47) ·100\% \qquad (9)$$

where ATR is the annual air temperature range calculated as the difference between the mean
temperatures of the warmest and coldest months, and $\varphi$ is the geographical latitude.

This index considers the dependence of ATR not only on geographical latitude, but also on the percentage of land in a given latitudinal band. The K index changes worldwide from –1.5% for areas with extreme oceanic climate to more than 140% for the extremely continental part of Eastern Siberia (Ewert, 1997). For more details about the methods used to calculate the
statistics mentioned above, see Przybylak et al. (2014, 2024).

The obtained results for the study period were compared against earlier temperature observations from the late 18[th] century (Przybylak et al., 2024) and with later records (Vinther et al., 2006), including modern observations from 1991–2020 (monthly data after Cappelen and Drost Jensen, 2021, and hourly data, based on which daily averages were calculated, after Drost
Jensen 2022). However, due to the lack of complete hourly data for the 4250 Nuuk (Fig. 1) station and the inability to calculate daily averages from 24 measurements for the whole period 1991-2020, the gaps in the daily averages were filled using an interpolation method (Nordli et al., 2020) based on data taken from the neighbouring 4254 Mittarfik Nuuk station (for more details see Przybylak et al., 2024, p. 1455).

Comparison of the same years 1807-13 was only possible with data taken from reanalyses: the NOAA/CIRES/DOE 20th Century Reanalysis (V3) (20CRv3) available for 1806–2015 (https://www.psl.noaa.gov/data/gridded/data.20thC_ReanV3.html, last_access_20



August 2025) (see also Slivinski et al. 2019, 2021) and the Modern Era Reanalysis for 1421-
2008 (ModE-RA; Valler et al., 2024) available via ClimeApp (Warren et al. 2024, https://mode-
ra.unibe.ch/climeapp/, last access 21 August 2025).

## 3. Results

### 3.1. Monthly resolution

Due to the irregularity of meteorological observations between 1 November 1806 and 31 July
1807, we present here an analysis for the period of the regular and complete series of
observations, i.e. August 1807 – July 1813 (6 full years). The average yearly temperature for
this period, calculated from August to July next year, reached -4.3 °C. The mean yearly
temperature in the coldest (1810/11) and warmest (1808/09) years reached -6.9 °C and -1.8 °C,
respectively (Table 2, Fig. 4). In the warmest of the six years, temperatures were higher than
the six-year average mainly from December to April, when mean monthly temperatures were
even greater than in the contemporary period 1991–2020. Similarly, in the coldest year, very
low winter and spring temperatures also accounted for the significant decrease in the annual
mean. On average, in the yearly cycle, the coldest temperature occurred in February (-14.4 °C)
and the warmest in July (5.3 °C) (Table 2, Fig. 4). In all analyzed years, the warmest month was
always July (except for the first studied year, when it was August), whereas the coldest month
occurred in a winter month or in March (Table 2). Year-to-year changes in mean monthly values
are clearly most significant in winter and spring months, oscillating from about 7 °C in January
and February to almost 10 °C in March. Conversely, the smallest year-to-year variations
(indicating the most stable months) occurred in October (0.5 °C) and September (1.1 °C),
whereas these variations were slightly greater in the summer months, ranging from 1.4 °C in
August to 2.7 °C in June (Table 2, Fig. 4).




**Table 2**. Mean monthly, seasonal and annual air temperature and variability (SD, DDTV) of MDAT in Nuuk in

the historical (1807–13) and contemporary (1991–2020) periods

| Period | Aug | Sep | Oct | Nov | Dec | Jan | Feb | Mar | Apr | May | Jun | Jul | SON | DJF | MAM | JJA | Aug-Jul |
|---|---|---|---|---|---|---|---|---|---|---|---|---|---|---|---|---|---|
| | Mean temperature (ºC) | | | | | | | | | | | | | | | | |
| 1807/08 | 3.1 | -1.6 | -2.2 | -4.0 | -6.6 | -19.4 | -12.9 | -10.7 | -6.3 | -2.2 | 0.6 | 1.5 | -2.6 | -13.0 | -6.4 | 1.7 | -5.0 |
| 1808/09 | 5.3 | -1.3 | -2.0 | -4.7 | -4.6 | -3.4 | -7.5 | -4.1 | -1.9 | -2.0 | -0.5 | 5.5 | -2.7 | -5.2 | -2.7 | 3.4 | -1.8 |
| 1809/10 | 4.1 | 1.2 | -1.2 | -2.5 | -9.3 | -8.3 | -9.2 | -5.6 | -4.3 | 1.8 | 3.8 | 6.6 | -0.8 | -8.9 | -2.7 | 4.8 | -1.9 |
| 1810/11 | 2.8 | -0.9 | -2.3 | -3.2 | -8.2 | -15.3 | -19.2 | -25.0 | -10.4 | -5.6 | 0.8 | 3.7 | -2.1 | -14.2 | -13.7 | 2.4 | -6.9 |
| 1811/12 | 4.9 | -0.2 | -2.2 | -9.0 | -10.4 | -11.7 | -25.3 | -13.5 | -4.6 | -1.0 | 3.0 | 7.4 | -3.8 | -15.8 | -6.4 | 5.1 | -5.2 |
| 1812/13 | 4.6 | -0.3 | -2.0 | -3.9 | -4.8 | -15.8 | -11.9 | -23.3 | -9.6 | -4.0 | 5.7 | 7.4 | -2.1 | -10.8 | -12.3 | 5.9 | -4.8 |
| **1807-13** | **4.1** | **-0.4** | **-2.0** | **-4.5** | **-7.3** | **-12.3** | **-14.4** | **-13.7** | **-6.2** | **-2.2** | **2.2** | **5.3** | **-2.3** | **-11.3** | **-7.4** | **3.9** | **-4.3** |
| **1991-2020** | **6.8** | **3.9** | **0.2** | **-3.2** | **-5.4** | **-7.4** | **-8.5** | **-7.6** | **-3.1** | **0.9** | **4.7** | **7.2** | **0.3** | **-7.1** | **-3.3** | **6.2** | **-1.0** |
| 1807-1813 – 1991-2020 (diff) | -2.7 | -4.3 | -2.2 | -1.3 | -1.9 | -4.9 | -5.9 | -6.1 | -3.1 | -3.1 | -2.5 | -1.9 | -2.6 | -4.2 | -4.1 | -2.3 | -3.3 |
| | SD (ºC) | | | | | | | | | | | | | | | | |
| Period | Aug | Sep | Oct | Nov | Dec | Jan | Feb | Mar | Apr | May | Jun | Jul | SON | DJF | MAM | JJA | Aug-Jul |
| 1807/08 | 3.5 | 2.7 | 3.2 | 2.8 | 3.3 | 7.1 | 8.4 | 5.7 | 7.2 | 3.3 | 3.2 | 3.0 | 2.9 | 6.2 | 5.4 | 3.3 | 4.4 |
| 1808/09 | 2.3 | 2.7 | 1.6 | 4.1 | 3.8 | 4.3 | 6.1 | 3.7 | 2.4 | 2.6 | 1.2 | 1.5 | 2.8 | 4.8 | 2.9 | 1.7 | 3.0 |
| 1809/10 | 1.5 | 2.1 | 2.2 | 3.9 | 3.4 | 5.1 | 6.6 | 6.2 | 4.4 | 2.9 | 2.2 | 2.5 | 2.7 | 5.0 | 4.5 | 2.1 | 3.6 |
| 1810/11 | 1.4 | 1.5 | 2.3 | 3.0 | 5.0 | 5.9 | 5.7 | 4.1 | 4.2 | 4.5 | 2.0 | 3.3 | 2.3 | 5.5 | 4.3 | 2.2 | 3.6 |
| 1811/12 | 2.9 | 2.2 | 2.5 | 2.8 | 5.0 | 4.8 | 8.7 | 9.1 | 5.0 | 2.4 | 3.1 | 2.5 | 2.5 | 6.2 | 5.5 | 2.8 | 4.2 |





| Period | Aug | Sep | Oct | Nov | Dec | Jan | Feb | Mar | Apr | May | Jun | Jul | SON | DJF | MAM | JJA | Aug-Jul |
|---|---|---|---|---|---|---|---|---|---|---|---|---|---|---|---|---|---|
| 1812/13 | 2.3 | 2.4 | 2.4 | 3.3 | 4.6 | 6.0 | 4.7 | 4.3 | 5.3 | 2.2 | 3.8 | 2.2 | 2.7 | 5.1 | 4.0 | 2.8 | 3.6 |
| **1807-13** | **2.3** | **2.3** | **2.3** | **3.3** | **4.2** | **5.6** | **6.7** | **5.5** | **4.8** | **3.0** | **2.6** | **2.5** | **2.6** | **5.5** | **4.4** | **2.5** | **3.8** |
| **1991-2020** | **2.1** | **1.9** | **2.7** | **3.7** | **4.8** | **5.3** | **7.2** | **5.8** | **4.3** | **3.6** | **2.5** | **2.0** | **2.8** | **5.8** | **4.6** | **2.2** | **3.8** |
| 1807-1813 – 1991-2020 (diff) | 0.2 | 0.4 | -0.4 | -0.4 | -0.6 | 0.3 | -0.5 | -0.3 | 0.5 | -0.6 | 0.1 | 0.5 | -0.2 | -0.3 | -0.2 | 0.3 | 0.0 |
| | | | | | | | | DDTV (°C) | | | | | | | | | |
| Period | Aug | Sep | Oct | Nov | Dec | Jan | Feb | Mar | Apr | May | Jun | Jul | SON | DJF | MAM | JJA | Aug-Jul |
| 1807/08 | 1.9 | 1.3 | 2.1 | 1.4 | 2.4 | 2.8 | 3.6 | 2.8 | 2.0 | 1.5 | 1.8 | 1.3 | 1.6 | 2.9 | 2.1 | 1.7 | 2.1 |
| 1808/09 | 1.4 | 1.5 | 1.1 | 1.9 | 2.4 | 2.3 | 3.4 | 2.4 | 1.6 | 2.0 | 1.1 | 1.2 | 1.5 | 2.7 | 2.0 | 1.3 | 1.9 |
| 1809/10 | 1.3 | 1.5 | 1.5 | 2.0 | 2.3 | 2.8 | 3.5 | 1.9 | 2.4 | 1.9 | 1.5 | 1.5 | 1.6 | 2.9 | 2.1 | 1.4 | 2.0 |
| *1810/11* | *1.3* | *1.0* | *1.6* | *2.4* | *2.9* | *3.2* | *2.5* | *2.3* | *2.5* | *2.0* | *1.5* | *1.9* | *1.6* | *2.9* | *2.3* | *1.6* | *2.1* |
| 1811/12 | 1.4 | 1.5 | 1.3 | 1.9 | 3.1 | 3.5 | 3.4 | 3.9 | 2.6 | 1.4 | 1.6 | 2.3 | 1.6 | 3.3 | 2.6 | 1.8 | 2.3 |
| 1812/13 | 1.2 | 1.2 | 1.2 | 1.8 | 2.5 | 2.8 | 3.3 | 2.2 | 4.0 | 1.9 | 2.3 | 2.0 | 1.4 | 2.9 | 2.7 | 1.8 | 2.2 |
| **1807-13** | **1.4** | **1.3** | **1.5** | **1.9** | **2.6** | **2.9** | **3.3** | **2.6** | **2.5** | **1.8** | **1.6** | **1.7** | **1.6** | **2.9** | **2.3** | **1.6** | **2.1** |
| **1991-2020** | **1.3** | **1.1** | **1.4** | **1.8** | **2.1** | **2.3** | **2.4** | **2.4** | **1.8** | **1.4** | **1.6** | **1.6** | **1.5** | **2.3** | **1.9** | **1.5** | **1.8** |
| 1807-1813 – 1991-2020 (diff) | 0.1 | 0.2 | 0.1 | 0.1 | 0.5 | 0.6 | 0.9 | 0.2 | 0.7 | 0.4 | 0.0 | 0.1 | 0.1 | 0.6 | 0.4 | 0.1 | 0.3 |

Key: SD – standard deviation, DDTV – day-to-day temperature variability, bold font – long-term average, underline – the warmest year, italic – the coldest year







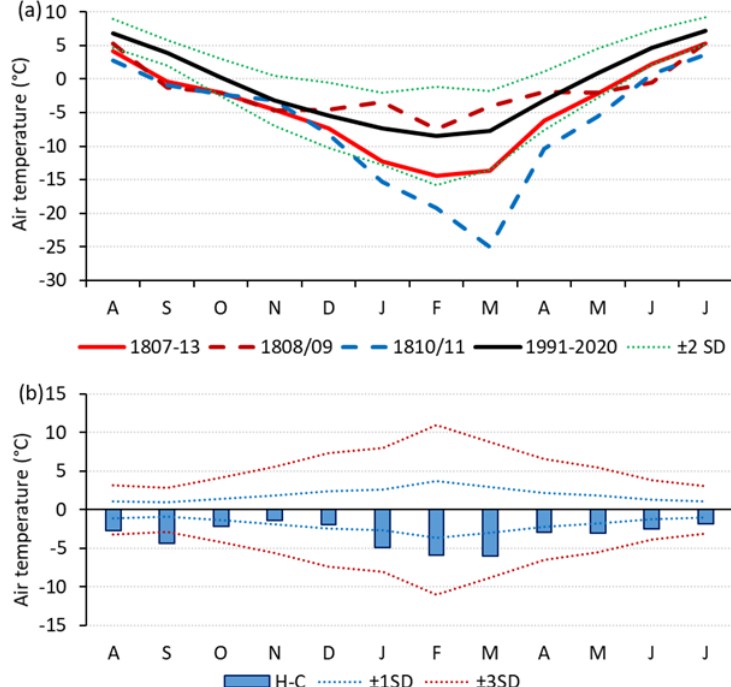

**Fig. 4.** a) Annual cycles of air temperature in Nuuk in the study historical period 1807–13, including the warmest (brown dashed line) and the coldest (blue dashed line) years, and in the contemporary period (1991–2020), b) differences between historical and contemporary mean monthly temperature values.

Key: SD values were calculated from the series of monthly mean temperatures from the contemporary period 1991–2020.

Reliable information about the thermal continentality of climate in the Arctic, calculated based on instrumental data, is unavailable for the period before 1840 because of the lack of long-term data series. For the first time, we have a six-year series of air temperature observations for Nuuk from the beginning of the 19th century. Based on data from Nuuk from 1991–2020, six-year moving averages of ATR and K were calculated. The calculation errors of these values compared to the 30-year period are small and amount to ±1.5 °C and ±3.5%, respectively. In the study period (1807–13), the mean ATR reached 24.0 °C, while K reached 49.2%. In the contemporary period, these values are 17.3 °C and 33.4%. This indicates that climate continentality in SW Greenland at the beginning of the 19th century was about 15% greater than at present. This difference is mainly accounted for by winter months having been significantly colder in the historical period than at present (see Fig. 4).





Statistics were also calculated for each observation time to roughly estimate the changes
in air temperature during the day. In line with expectations, the average annual temperature was
highest at midday (-2.1 °C) and lowest in the morning (-4.8 °C) (Table S1). At all times of the
day, the warmest month was July, while the coldest was February (midday and morning) or
March (evening). In the study period, the highest temperature (18.5 °C) was noted in midday
on 16 July 1810, while the lowest (-36.5 °C) occurred in the morning and evening observation
times on 21–24 February 1812. Thus, the absolute range of temperature reached as much as
55 °C. The temperature in the historical period in all months was lower than today, in particular
in February and March, when the difference reached about 6 °C (Table 2, Fig. 4). On the other
hand, this difference was the smallest in November (1.3 °C), as well as in December and July
(1.9 °C in both). On average, according to annual averages, the historical period was 3.3 °C
colder than today (1991–2020). The warmest (1808/09) and coldest (1810/11) years were also
colder than today – by 0.8 °C and 6.9 °C, respectively (Table 2, Fig. 4).

3.2. Daily resolution

Most studies analyzing climate and its changes in various areas of the globe, including the polar
regions (including the Arctic), most often use monthly, seasonal and annual data. For many
purposes, these are overly generalized data that do not allow for a complete understanding of
the wide range of climate features that characterize each area, particularly the extreme
phenomena. Therefore, over the last 20 years, we have started using higher-resolution data, i.e.
daily and sub-daily, for our analyses of weather and climate in the Arctic (e.g., Przybylak and
Vizi 2005; Nordli et al. 2014, 2020; Przybylak et al. 2016, 2018, 2022, 2024; Przybylak and
Wyszyński, 2017). That is why we also present here some statistics based on this kind of data,
and, thanks to that, we present an in-depth analysis of the weather and climate in Nuuk during
the study period.

Figure 5 presents the annual cycle of air temperature in Nuuk based on MDAT for each
study year and the average for the six-year period 1807–13. On average, with few exceptions,
MDATs were colder in the historical period than at present. The greatest differences occurred
in winter months (Dec–Mar) and September, while the smallest occurred from mid-May to mid-
August. For some of the historical years, the picture is different; specifically, some had a greater
frequency of MDATs that were warmer than today than did other years. Nonetheless, these of
course occurred more rarely than did days that were colder in the past than at present. The
MDATs that were most similar to present conditions were noted in the warmest year, i.e.



1808/09, while MDATs were least similar in the coldest year (1810/11) (Fig. 5). The year
1808/09 was the warmest due to the very high MDATs from mid-December to mid-April, which
were significantly greater than present MDATs. Conversely, the year 1810/11 was the coldest
due to a significant, prolonged period of very low MDATs from the beginning of January to the
start of April. This means that MDAT values in winter and the first half of spring are mainly
responsible for the mean annual air temperature values, which is a typical pattern for moderate
latitudes, but particularly for polar regions (Przybylak, 2016). MDATs in the historical period
usually do not exceed 2 SD from the contemporary long-term average MDAT (Fig. 5).





**Fig. 5.** Annual courses of MDAT in Nuuk in historical years (lines in different colours) and 1991–2020
mean (black line). Blue (±1 SD) and green (±2 SD) dashed lines indicate SD calculated using MDAT from
the period 1991–2020 and are added and subtracted from the 30-year mean.






More precise information about the character of MDAT changes between historical and present times is contained in Fig. 6, which shows relative seasonal frequencies of occurrence of specific MDAT values stratified into one-degree intervals. In all seasons, warmer MDAT values occur clearly more frequently in the contemporary period than in the historical period. The

range of the highest-frequency MDAT values frequencies also shifted towards higher values for all seasons. For example, for winters, they shifted from being flatly distributed across a wide range (-16 °C to -5 °C) to a more narrowly spread range (-10 °C to -5 °C) comprising with higher frequencies of MDATs (Fig. 6). The shape of MDAT distributions in all seasons and in both historical and contemporary periods are close to normal because both skewness ($\gamma1$) and

kurtosis ($\gamma2$) are not greater/smaller than $\pm1$, except $\gamma1$ in spring in the historical period (1.03). The left-skewed distribution of MDATs is a very characteristic feature of the climate in all seasons in Greenland, except for summer in the contemporary period. Leptokurtic distribution is also a very characteristic feature of MDAT distribution in all seasons (except summer) in the historical period, whereas, in modern times, weak platykurtic distribution ($\gamma2>-0.2$) is evident

except in spring (Fig. 6).

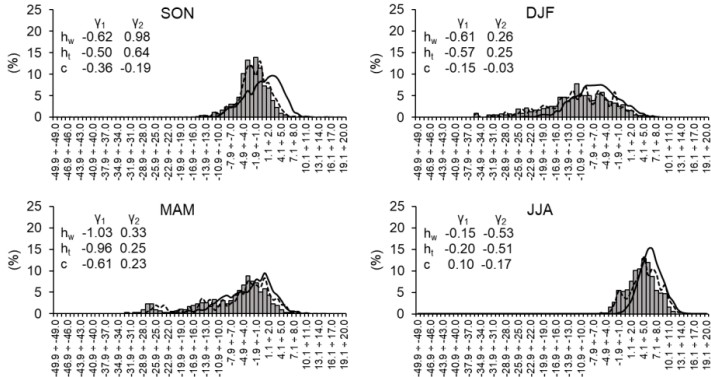

**Fig. 6.** Seasonal (SON, DJF, etc.) relative frequencies of occurrence (in %) of MDAT in historical (bars – weighted mean, dashed lines – simple arithmetic means from three measurement times) and modern (solid lines) sites located in Nuuk 1807–13 and 1991–2020. Values of skewness ($\gamma1$) and kurtosis ($\gamma2$) for historical weighted mean ($h_w$),

simple arithmetical mean ($h_t$) and contemporary (c) times are also shown.

Annual courses of DDTV in Nuuk are shown for each of the six historical years for which complete or near-complete data are available for the entire year (Table 3, Fig. 7). Similarly to the climate of the present (Przybylak, 2002) and to the late 18[th] century (Przybylak et al., 2024), the DDTV in the study period was greatest in the cold half-year, but particularly



in winter, and lowest usually from May to October, but especially in summer (Fig. 7). In summer, values of DDTV are approximately similar among all historical years, usually not exceeding 5.0 °C. Only on single days do values of DDTV oscillate between 6 and 8 °C. In winter, changes in DDTV are greater than in summer, and their values are also several times greater. In each year except the warmest year (1808/09), there are some spans in which DDTVs

exceed 10 °C, but they are less than 15 °C, except one value at the end of February 1812 (see Fig. 7). The change in measured temperature from evening 25 February (orig. 36 °K) to morning of 26 February (orig. 1.5 °W) reached 37.5 °C (MDAT difference 35.6 °C). Under the table of data from February 1812, Giesecke wrote the following (see Fig. 3): "Hochste Kälte 21sten - 23sten, 24sten und 25sten [Februar] 36 ½ °. Wärmer den 26ten – 1 ½ °" [translation, first author:

Extremely cold 21$^{st}$ – 23$^{rd}$, 24$^{th}$ and 25$^{th}$ [February] -36.5 °C. Warmer the 26th − 1.5 °C.). February 1812 was exceptionally cold according to the Giesecke measurements. The temperature between 5 and 25 February never rose above -20 °C, and between 19 and 25 February it was even above -30 °C (see also Figs 3 and 5). In the days mentioned by Giesecke in his written note (21–25 Feb), the temperature oscillated constantly between -34 °C and

-36.5 °C (see Fig. 3).

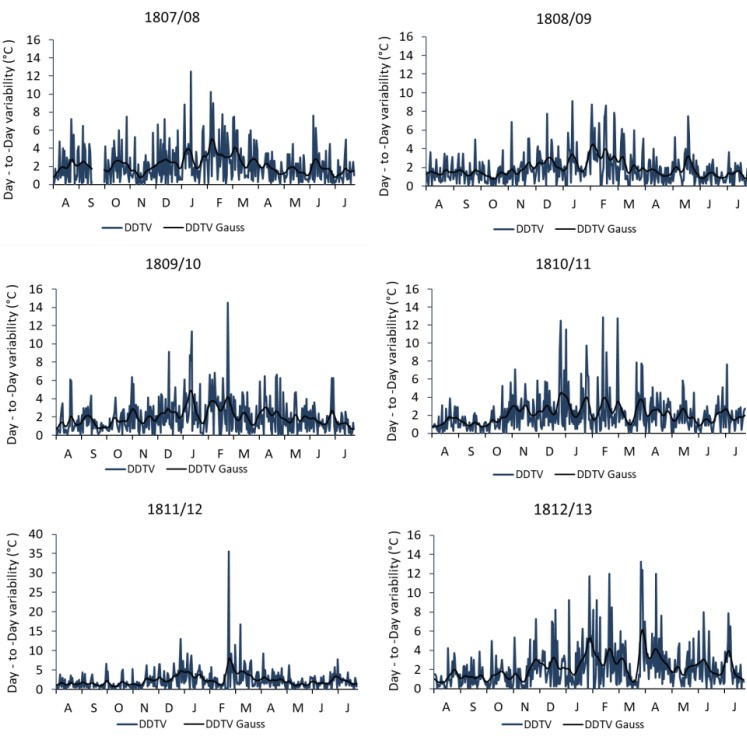



**Fig. 7.** Annual courses of DDTV in Nuuk in historical periods. Individual days (grey) are filtered by a Gaussian low-pass filter (black) with a standard deviation of 3 d in its distribution, corresponding to a rectangular filter of about 10 d.

*In February 1812 the shift from –35.6 °C to 0 °C produced an exceptional DDTV of 35.6 °C, requiring the vertical axis to extend to 40 °C to capture the full variability

The question arises: What could be the reason for such great heating from one day to the next? From the meteorological register, we know that, from the evening of 25 February to

the morning of 26 February, the wind direction changed from NE to SE (see Fig. 3). From Fig. 9, it is evident that, in Nuuk in the historical period in winter, such a change of wind direction brought an average warming of about 14 °C. We can only speculate that, during the night, a very strong fohn may have occurred from the area of the Greenland ice sheet, significantly enhancing the effect associated with the mentioned change in direction of air mass inflow.

To compare the DDTV shown in Fig. 7 against the analogous data from contemporary years, two thermally contrasted years from the period 1991–2020 were selected (i.e., the warmest [2018/19] and the coldest [1992/93]). The results presented in Fig. S1 show that the scale of differences between these two years and years taken from the historical period is similar. The average yearly differences in DDTV between years 2018/19 and 1992/93 reached

only 0.3 °C, but the highest were in the first year. Therefore, a more detailed comparison of the DDTV in the historical period is shown only for the year 2018/19 (Fig. 8).

The averaged annual DDTVs in Nuuk, during the studied historical years, were usually slightly greater (by 0.1 to 0.5 °C) than in 2018/19. Looking at the annual cycle, it is evident that the greatest positive differences were usually noted between September and March, and the

smallest occurred in the rest of the year (see Fig. 8). Such changes in DDTVs are in line with expectations, which are that, in a warmer climate, a decrease in high-frequency temperature variability should be observed (e.g., Karl et al. 1995; Zwiers and Kharin, 1998; Przybylak, 2002).





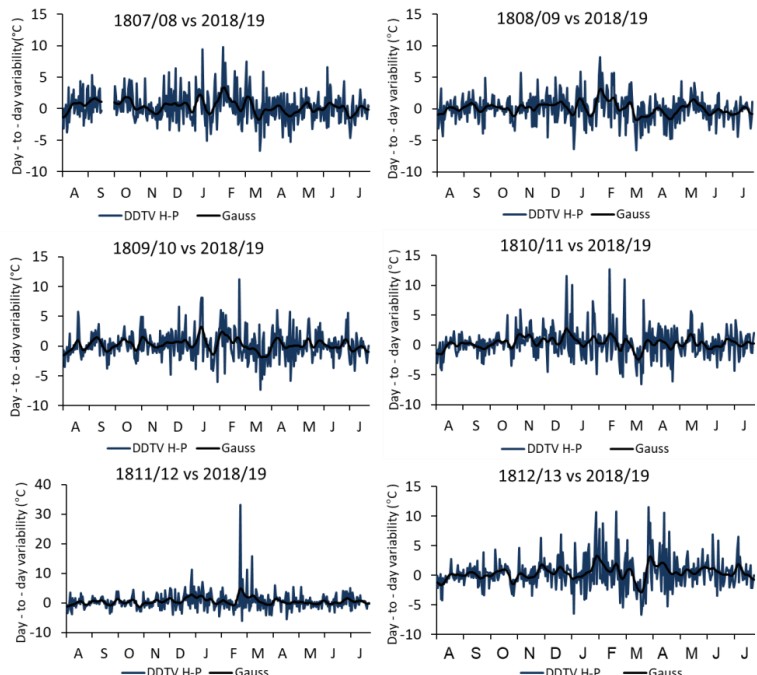

**Fig. 8.** Differences in DDTV in Nuuk between historical and contemporary (2018/19) periods. Individual daily differences (grey) are filtered by a Gaussian low-pass filter (black) with a standard deviation of 3 d in its distribution, corresponding to a rectangular filter of about 10 d.

*In February 1812 the shift from –35.6 °C to 0 °C produced an exceptional DDTV of 35.6 °C, requiring the vertical axis to extend to 40 °C to capture the full variability.

The analysed series of observations available for the years 1807–13 also contains observations of wind directions. Therefore, we decided to calculate how, on average, wind direction influenced the value of MDATs both in historical and contemporary periods. For this purpose, we use "thermal roses" consisting of eight wind directions. They were constructed for all seasons and for the year (Fig. 9). It is clear that, at present, wind direction influences air temperature in Nuuk significantly less than it did in historical times; the roses are more regular. In historical times, the irregularity of thermal roses is especially great in winter and spring. From historical to present times, the greatest thermal conditions in winter and spring changed for the winds coming from the sector N–NE and also E in winter and NW in spring (Fig. 7). The explanation of these changes is unknown. However, it is possible that one reason is connected with the recession of the Greenland ice sheet and local icefields in the last 200 years surrounding Nuuk from the northern sector. In all seasons, winds from all directions bring



warmer air masses at present than in the historical period, except for winds from some southern directions in winter (Fig. 7).

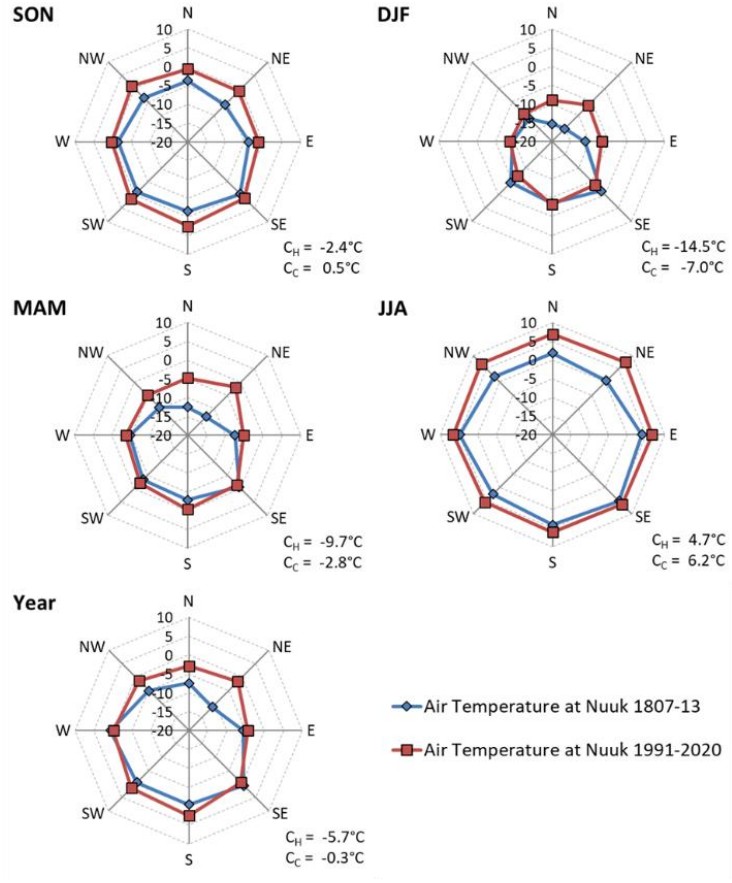

**Fig. 9.** Seasonal and yearly thermal roses of wind in Nuuk in the historical (1807–13) and contemporary (1991–2020) periods

At the moderate latitudes, to distinguish the thermal seasons (winter, spring, summer, and autumn), three thresholds are usually used, i.e. 0 °C, 5 °C, and 15 °C. For the Arctic, including Greenland, such thresholds are not appropriate because, as Przybylak et al. (2024) write, "the annual cycle is significantly flatter and less clear than at moderate latitudes and is dominated by the winter, which is much longer than the other seasons" (Przybylak 2016). For this reason, for the polar regions, Baranowski (1968) proposed other criteria to delimit the four standard seasons using the thresholds -2.5 °C, 0 °C and 2.5 °C (for details, see Przybylak et al. 2024). Using this division of the year into seasons has the additional advantage that, in addition




to providing information about the value of the average temperature or other weather elements, it makes it possible to study changes in the durations of seasons, as well as their changes in dates of onset and ending. All this information is very important for people's everyday lives and also, for example, for agricultural, economic and tourist activities.

In the historical time, the longest season was winter, varying from 161 days in 1809/10 to 221 days (1810/11), and the shortest was spring (from 32 to 85 days) (Fig. 10). On average, winter lasted 191 days, i.e. more than half of the year. The durations of the other seasons were roughly equal to one another, at 50 days for spring to 64 days for summer. From historical to contemporary times, the duration of summer has increased markedly – on average, by 58 days

(Fig. 10). The other seasons showed decreases in duration, ranging from 13 days for spring to 27 days for winter, i.e. each of less than half the scale of the increase in summer duration.

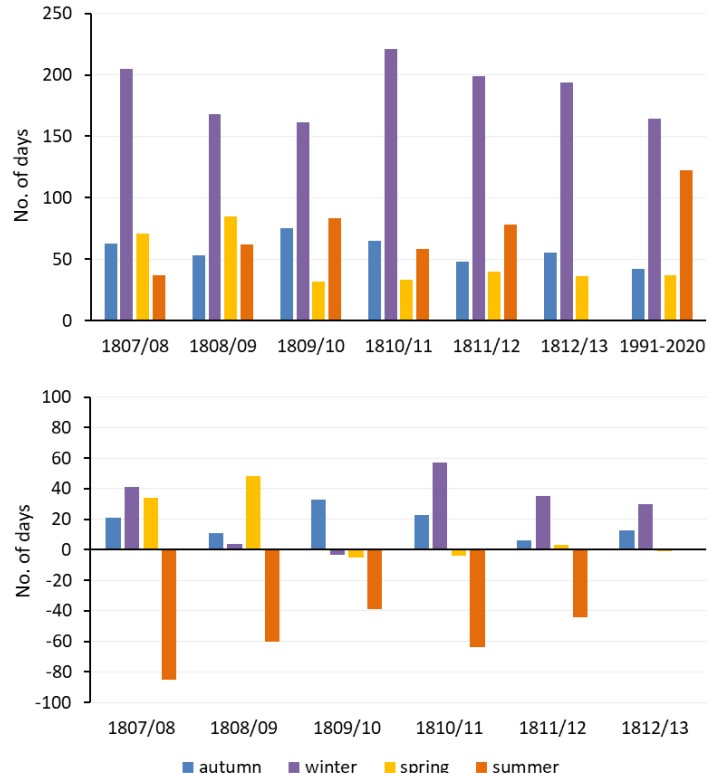

**Fig. 10.** Duration of thermal seasons in Nuuk in the historical (1807–13) and modern (1991–2020) periods (upper figure) and their differences (lower figure). Key: contemporary data were subtracted from historical data.



## 4. Discussion and conclusions

This newly discovered series of meteorological measurements in Nuuk (SW Greenland), encompassing a complete series of observations from Aug 1807 to July 1813, is very unique and therefore very valuable for the study of climate in this part of the Arctic. Analyses of the history of early instrumental meteorological observations, conducted by Przybylak et al. (2010) for the entire Arctic and by Vinther et al. (2006) for Greenland, lead to the conclusion that the newly discovered series is the longest which exists for the Arctic prior to the 1840s. Its usefulness, apart from providing detailed insight into the climatic conditions in SW Greenland at the beginning of the 19<sup>th</sup> century, is much broader. The data can be used for the calibration of, among others: (i) climate reconstructions based on various proxy data; (ii) climate reconstructions based on models; and (iii) climate reconstructions from the NOAA/CIRES/DOE 20th Century Reanalysis (V3) (20CRv3) available for 1806–2015 (see also Slivinski et al. 2019, 2021). Slivinski et al. (2021) documented that the 20CRv3 is a good tool for reconstructing even extreme weather and climate events, and we therefore plan to use this reanalysis to study (in a separate paper) the reason for the described great warming from 25 to 26 February 1812.

Better knowledge about the climate in Greenland at the beginning of the 19<sup>th</sup> century can help explain the reasons for very cold spells that have occurred on this island in the past millennium. We documented in the Results section that the temperature in the study period was considerably lower than in the contemporary period (1991–2020). We also checked how this period compares thermally with other available instrumental data (the newest and the oldest). For this purpose, we calculated 30-year average temperatures (Aug 1870–Jul 1900, Aug 1900–Jul 1930, …, Aug 1990–Jul 2020), starting from the beginning of the available continuous series of meteorological measurements in Nuuk since 1866 (Przybylak, 2000; Vinther et al., 2006) (Fig. 11). Moreover, in order to have periods that are more comparable in length with the study period, we also chose, from the entire mentioned series (1870–2020), two six-year periods representing the periods that were the warmest (1926–32) and the coldest (1881–87) and simultaneously described the range of thermal changes in 1870–2020 (Fig. 12a). For Nuuk, for the period before 1870, a complete five-year series of temperature data (1816–20) is also available that was gathered by Vinther et al. (2006). We used this series for comparison against the thermal state of the period 1807–1813. In addition, we also added a curve to Fig. 12a that represents the mean yearly cycle of temperature based on data available for the period 1784–92 (after Przybylak et al. 2024). How the available datasets from reanalyses (20CRv3 and ModeE-



RA) reconstruct the surface air temperature is shown in Fig. 12b. Figs. 11 and 12 clearly indicate
that the period under study was extremely cold. It was significantly colder than: i) all 30-year
periods; ii) the coldest six-year period in the series 1871–2020; iii) the late 18[th] century; and
iv) reconstructed temperature for the period 1807-13 by 20CRv3. The period 1807–13 was also
colder (but only slightly) than the five-year period 1816–20. Only the ModeE-RA data are
significantly colder than the measured data from October to February, while for the remaining
490 months they are similar (Fig. 12b). Vinther et al. (2006) concluded that the decade 1810 was
the coldest in the entire record available for Nuuk, i.e. since 1784. They have at their disposal
data from the mentioned five-period as well as for the years 1811 and 1812. The data come,
however, from another locality than the place of meteorological observations made by
Giesecke. A comparison of data for the latter two years (1811 and 1812) presented by Vinther
et al. (2006) and in this study is shown in Fig. S2. There is a good coherence between monthly
means from June to December, but in the rest of the year, the differences are quite large. The
reasons for these are difficult to explain, but they may be connected mainly with the localities
of measurement places being different. Probably, the data presented by Vinther et al. (2006)
comes from a site located near the coast, while Giesecke's site is assumed to have been located
at some distance from the coast and probably also at a higher altitude. Brohan et al. (2010)
studied Arctic marine data from the seas surrounding the south part of Greenland in the period
1810–25 and found that this period was also "significantly cooler than that of today".

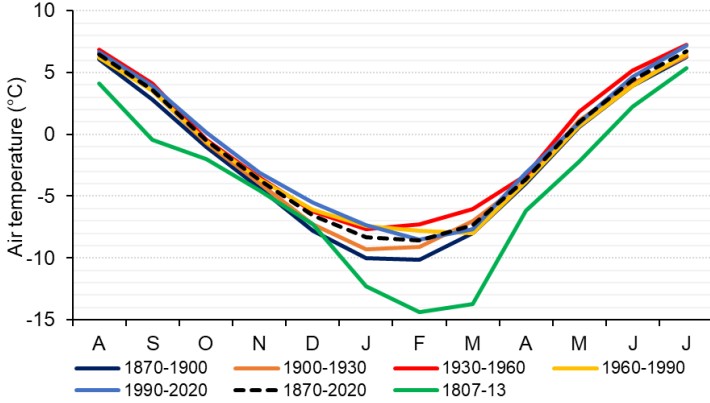

**Fig. 11.** Yearly cycle of mean air temperature in Nuuk in the period 1807–13 compared to the mean from the entire
series (1870–2020) and means calculated from that series for 30-year periods (data taken from Vinther et al. 2006).



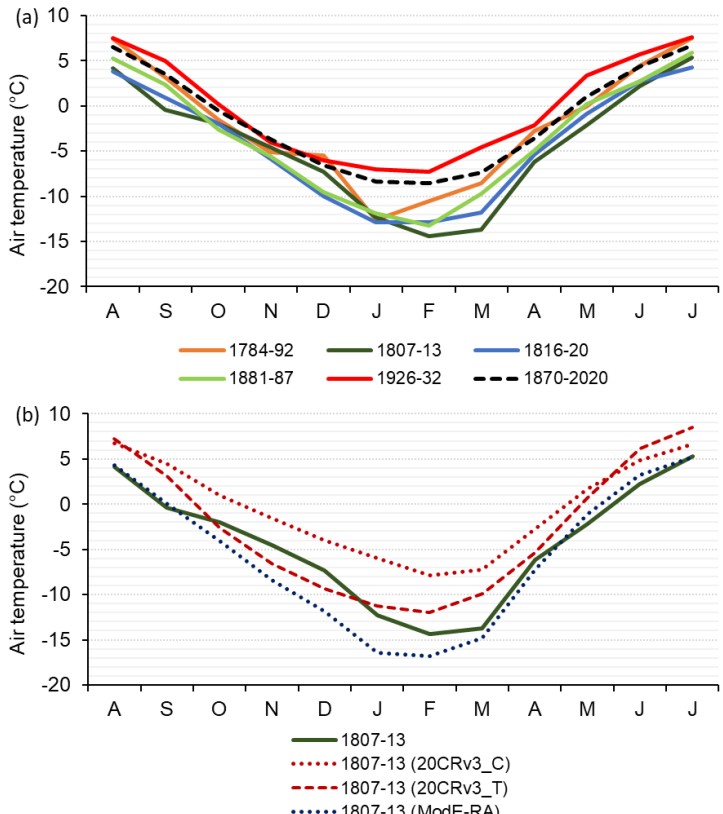

**Fig. 12.** Yearly cycle of mean air temperature in Nuuk in the period 1807–13 compared to: a) the mean for 1870–2020 and for various short-term series available since the late 18th century. Explanation: 1784–92 – data after Przybylak et al. (2024); 1816–20 – data after Vinther et al. (2006); Jan 1870–Jul 1990 – data after Vinther et al. (2006); Aug 1990 – 2020 data after Cappelen and Drost Jensen, 2021, and b) the mean for 1807-13 taken from reanalyses: 20CRv3 (C – coastal grid point, T – terrestrial grid point; Slivinski et al., 2019) and ModE-RA (Valler et al., 2024)

The question arises as to whether and to what extent this cold period in SW Greenland at the beginning of the 19$^{th}$ century is correctly reflected in air temperature reconstructions based on proxy data and computer climate simulations. Unfortunately, in the area of Nuuk, there does not exist any series of proxy data of value for the reconstruction of air temperature; the closest according to Kaufmann et al. (2009) and McKay and Kaufmann (2014) is located in Lake Braya Sø, lying about 313 km to the north (φ= 67° 00' N, λ= 50° 42' W). Lake sediments allow for reconstructing summer temperature only, according to Table 1 in McKay and





Kaufman (2014), but the reconstruction is not shown. Further north in the Ilulissat region, there are other lakes for which proxy data are available, and it is possible to reconstruct July temperature based on chironomids present in sediments (Axford et al., 2013), which were not included in the paper written by McKay and Kaufman (2014). The temperature reconstructions presented in Axford et al. (2013) are not of sufficient resolution to be used for comparing short-term temperature changes in the last millennium. Most of the proxy data for Greenland (and the most well-known) comes from the Greenland Ice Sheet. According to the information given in Table 1 (McKay and Kaufman, 2014), ten series of proxy data exist in a PAGES Arctic 2k database published in 2013 by PAGES 2k Consortium. All of them use ice-core data ($\delta^{18}$O) and allow for the reconstruction of annual mean temperatures. Recently, two series of annual air temperatures for central (Kobashi et al., 2010) and central-northern (Hörhold et al., 2022) Greenland for the last millennium were reconstructed using ice-core data, i.e. isotope ratios of nitrogen (15N/14N) and argon (40Ar/36Ar) in air bubbles and $\delta^{18}$O, respectively. According to Kobashi et al. (2010), the beginning of the 19[th] century saw one of the coldest temperatures in the last millennium; lower temperatures were noted only in two short periods in the 18[th] century and in the first half of the 17[th] century. The reconstruction presented in Fig. 1 of Hörhold et al. (2022) shows a picture for central-northern Greenland that differs only slightly. It shows four periods of similar size cooling in the last millennium, including one in the first half of the 19[th] century. It also shows that the mean annual air temperature at the beginning of the 19[th] century was about 3 °C colder than the present time and about 2 °C colder than medieval times. Using instrumental data from Nuuk from the periods 1807–13 and 1991–2020, we found a 3.3 °C increase in annual air temperature from historical to present times. This indicates a very good agreement between the measured and temperatures reconstructed using ice-core data. As we mentioned earlier, such a good correlation (r=0.67 and r=0.60 for the periods 1785–1872 and 1873–1970, respectively) between the extended SW Greenland temperature series and the ice-core was also found by Vinther et al. (2006). Hörhold et al. (2022) found that the Arctic 2k reconstruction after McKay and Kaufman (2014), which represents large parts of the higher Arctic circumpolar region, shows a low correlation over Greenland. Nevertheless, the greatest cooling in the last millennium, according to the Arctic 2k reconstruction, evidently occurred at the beginning of the 19[th] century. This is particularly clear for the revised PAGES 2k Arctic reconstruction (see Fig. 2 in McKay and Kaufman, 2014). According to that reconstruction, the mean annual Arctic air temperature at the beginning of the 19[th] century was about 2 °C colder than at present. Arctic-wide summer-weighted annual temperature reconstructed for the last 400 years by Overpeck et al. (1997, see their Fig. 3) also shows that a great cooling started at the



beginning of the 19th century but peaked in the 1840s. The temperature at this time was about
       1.5 °C colder than in the late 20th century. Also, the spatially resolved two-millennium summer
       (Jun–Aug) temperature reconstructed for the Arctic and sub-Arctic domain (north of 60°N)
       calculated by Miller et al. (2018) reveals a great cooling in the first half of the 19th century,
       being one of the three coolest episodes in the two millennia analysed. In comparison to the

present, the temperature in the first half of the 19th century was about 2 °C colder (see their Fig.
       2). This reconstruction is very close to the reconstruction of mean annual air temperatures
       presented by McKay and Kaufman (2014); the latter, however, shows a cooling that was greater
       and occurred earlier than in the case of reconstructed summer temperatures (see Fig. 3 in Miller
       et al. 2018). A two-millennium summer (Jun–Aug) temperature reconstruction over the Arctic

and sub-Arctic domain (north of 60°N) is also presented by Werner et al. (2018). They used a
       set of 44 annually dated temperature-sensitive proxy archives of various types from the PAGES
       2k database, revised in 2017. Also, in this reconstruction, a cooling at the beginning of the 19th
       century is evident that is the greatest of the entire 2000 years. Werner et al. (2018) compared
       their reconstruction with the other six reconstructions published in recent years and found their

reconstruction to be very close to that presented in the work of McKay and Kaufman (2014).
       The summer cooling, however, according to this reconstruction (see Fig. 3 in Werner et al.
       2018), was about 0.5 °C smaller than McKay and Kaufman's reconstruction. Other
       reconstructions presented by Werner et al. (2018) in their Fig. 3 (i.e., Kaufman et al., 2009; Shi
       et al., 2012; Hanhijärvi et al., 2013; Tingley and Huybers, 2013) also confirm the occurrence of

significant cooling, except the last-cited paper. In conclusion, we can state that the cooling in
       the first half of the 19th century was common throughout the Arctic, including Greenland.

            Let us now check how climate models reconstruct this period in Greenland and the
       Arctic. According to the annual mean surface temperature reconstructed for the Arctic using an
       ensemble of five simulations performed with a global three-dimensional, atmosphere–sea-ice–

ocean model driven by both natural (solar and volcanic) and anthropogenic (increase in
       greenhouse gas concentrations and tropospheric aerosols) forcings, the last millennium featured
       four periods of great cooling. These were in the mid-11th century, the second halves of the 15th
       and the 17th centuries, and the beginning of the 19th century (see Fig. 1c in Goose and Renssen,
       2003). The last two cooling periods simulated by the model occurred in the years 1670–1700

and 1800–30, when surface temperatures were about 0.5 °C colder than the mean in the Arctic
       from the period 1000–1850. Another modelling work, presented by Crespin et al. (2012), using
       the three-dimensional Earth system model of intermediate complexity LOVECLIM forced by
       changes in solar irradiance, volcanic activity, land use, greenhouse gas concentrations and



orbital parameters, also shows a very cold period in the first half of the 19th century. This was the coldest period in the entire last millennium. The temperature difference in comparison to present times reached about 1.8 °C. Figure 1.4 in Crespin (2014) shows that simulations of annual mean Arctic temperature in five different GCM models (CCSM4 [USA], GISS-E2-R [USA], IPSL-CM5A-LR [France], MPIESM-P [Germany], and Bcc-csm1-1 [China]) confirm the existence of cold period in the first half of the 19th century, but this was not always found

to be the coldest period in the past millennium.

According to investigations conducted by Crespin et al. (2012) and Crespin (2014), the reasons for this great cold period at the beginning of the 19th century were mainly intense volcanic activity, in particular in the period 1810–40, and a smaller-scale decrease in solar irradiance connected with the Dalton minimum occurring in 1790–1830 (most intensely in

1797–1827; Silverman and Hayakawa, 2021) as well as changes in land use. Volcanic eruptions were the main reason for the cooling that occurred in the 1810s (in particular, in 1811 and 1817–18), according to Vinther et al. (2006). They connected these two cold spans with eruptions of an unidentified volcano around 1809 and the very well-known eruption of the Tambora volcano in 1815, respectively. Box (2002), investigating air temperature variability in Greenland in

1873–2001, found that the greatest anomalies in this time were also connected with large volcanic eruptions, sea-ice extent and the NAO. Brohan et al. (2010), analysing marine data extracted from the reports of the noted whaling captain William Scoresby Jr (Greenland Sea) and from the records of a series of Royal Navy expeditions to the Arctic preserved in the UK National Archives, concluded that the cold early-19th-century climate was linked with low solar

activity and followed a series of large volcanic eruptions.

Summarising the obtained results, we can again underline that the first two decades of the 19th century in the study region, in all of Greenland and on average in the Arctic, were one of the coldest periods in the past two millennia (and possibly the coldest). It is also probable that such a large scale of cooling and its durability are one of the main reasons that almost all

available reconstructions of air temperatures using different proxy data or using climate models for this purpose are almost fully consistent with the available meteorological observations for this period. For example, such good correlations between instrumental temperature data and different reconstructions are not observed for analyses encompassing the earlier period (1784–92) for SW Greenland (for details, see Przybylak et al., 2024). The late 18th century in SW

Greenland and many places in the Arctic was warm, but the warming in this time was not so spectacular as the cooling that occurred in the Arctic at the beginning of the 19th century. Also, a good agreement exists concerning the reasons for the cooling of the early 19th century in



Greenland and the Arctic. The two main reasons underlined by most authors studying this issue are mainly intense volcanic activity and, to a lesser degree, low solar activity connected with the Dalton minimum. In addition, land use changes in mid-latitudes are also mentioned by some researchers. Changes in atmospheric circulations described using the NAO must also be taken into account in the case of SW Greenland's climate.

There are many advantages of instrumental observations in comparison to the possible reconstruction of climate using proxy data and simulations by climate models. Reconstructions are less precise, have smaller resolution and are limited most often to i) only some parts of the year (mainly to summer) and ii) one main climate variable (air temperature). For this reason, searching for new meteorological data series in archives and libraries is extremely important and necessary. It seems that it should still be intensified despite the fact that, for the last approximately 30–40 years, this type of activity has been carried out intensely worldwide. The newly discovered unique series of meteorological observations for SW Greenland for the period 1807–13 that is presented in this article proves that new data of this type can still be found for various parts of the globe, including the Arctic, where they are many times more valuable than comparable data for moderate latitudes (for which the network of stations is significantly denser). Unfortunately, fewer scientists are involved in this type of data rescue activity for the Arctic at present than in previous times. Most scientists are focused on climate reconstructions based on proxy data and climate model simulations, which is very good, but we should also motivate young scientists to be more interested in the history of the Arctic climate based on early instrumental observations and other documentary evidence.

**Data availability.** Datasets of surface air temperature for this research were derived from the following public domain resources:
1. Repository for Open Data (RepOD), Nicolaus Copernicus University Centre for Climate Change Research collection, https://doi.org/10.18150/IGYNGV, as cited in Przybylak et al. (2025),
2. Vinther et al. (2006) and Danish Meteorological Institute (DMI); https://www.dmi.dk/publikationer/, as cited in Cappelen and Drost Jensen (2021) and Drost Jensen (2022),
3. Repository for Open Data (RepOD), Nicolaus Copernicus University Centre for Climate Change Research collection, https://doi.org/10.18150/L1Y21Q as cited in Singh et al. (2023),
4. The NOAA/CIRES/DOE 20th Century Reanalysis (V3) (20CRv3), https://www.psl.noaa.gov/data/gridded/data.20thC_ReanV3.html (Slivinski et al. 2019)
5. The Modern Era Reanalysis (ModE-RA, Valler et al. 2024) via ClimeApp https://mode-ra.unibe.ch/climeapp/ (Warren et al. 2024)



**Acknowledgements.** We express our gratitude to the local Inuit communities on the south-western coast of Greenland, who probably cooperated with Sir Charles Lewis Giesecke in their lands in the 19th century. Support for the Twentieth Century Reanalysis Project version 3 dataset is provided by the U.S. Department of Energy, Office of Science Biological and Environmental
Research (BER), by the National Oceanic and Atmospheric Administration Climate Program Office, and by the NOAA Earth System Research Laboratory Physical Sciences Laboratory.

**Financial support.** This research has been supported by the National Science Centre, Poland (grant-no. 2020/39/B/ST10/00653)


**Author contributions**. Study design by RP. Data collection and selection by AA, PW, and RP. Data curation by AA, GS, KCh, PW, and RP. Literature review by RP. Statistical analysis and visualisation by AA, GS, KCh, PW and RP. Interpretation of results by RP, PW, and AA. RP developed the first draft of the manuscript. All the authors contributed to reviewing and editing
the manuscript. RP and PW contributed to editing and approved the final version of the manuscript

**Competing interests.** The contact author has declared that none of the authors has any competing interests.

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
