# Peer review of "Newly discovered series of meteorological measurements in SW Greenland (Nuuk) in the period 1806–13"

_EGUsphere, 2025_

## Author Comment (AC2)

**Reply to Reviewer 2** (in green font)

The manuscript analyzes what the authors describe as a "newly discovered series of historical meteorological data" recorded by Karl Ludwig Giesecke in Nuuk between 1806 and 1813, with regular measurements from 1807 to 1813. The authors claim to compare Giesecke's data with recent observations from the same location.

The paper outlines what the authors call the "state of knowledge about early meteorological observations" and formulates two main objectives:

To present this newly discovered series of observations to a broader scientific audience and to offer preliminary results of a climate analysis—limited to a description of conditions and changes in air temperature in Nuuk.

To compare air temperatures during the study period with earlier measurements from the late 18th century as well as later records, including modern observations from 1991–2020.

The authors identify the place of measurement as "the territory of present-day Nuuk." They briefly introduce the author of the manuscript, though the title, language, and signature are not mentioned, and the archival location is initially described only as "Royal Society," with further details provided later. Throughout the paper, the authors repeatedly speculate about the authorship of the text and the exact location of the thermometer.

Reply. Thank you for these comments and suggestions. All additional information has been added, including a more detailed description of the source used. The following texts were added:

*Currently, they are available in manuscript MA/154 in the archives of the Royal Society in London. … The title of the manuscript is: Meteorological observations at Godthåb [Nuuk], Greenland, by Charles Lewis Giesecke. More details about this source are available in https://catalogues.royalsociety.org/CalmView/Record.aspx?src=CalmView.Catalog&id=MA%2f154&pos=1.*

We do not speculate about the authorship of the text (of the source) but about the need to utilise the local population for meteorological observations, likely including the Moravian Brethren residing there, who had experience in conducting such observations. This applies to the periods when Giesecke travelled along the SW Greenland coast. We do not know the exact location within the then-settlement of Godthaab where Giesecke conducted his meteorological observations, but this does not significantly impact the obtained results, as air temperatures in the Arctic correlate well, even over much greater distances (see Przybylak 1997, 2002).

Przybylak R., 1997, Spatial variation of air temperature in the Arctic in 1951-1990, Polish Polar Research, 18, 1, 41-63.

Przybylak R. 2002, Variability of air temperature and atmospheric precipitation in the Arctic, Atmospheric and Oceanographic Sciences Library, 25, Kluwer Academic Publishers, Dordrecht/Boston/London, pp. 330.

Evaluation

The manuscript lacks a solid historical methodology for working with handwritten German-language sources. Consequently, the object of study is not clearly defined, and historical methods for analyzing early measurements and observations are not adequately applied.

Reply. We would like to point out that, in this article, we used only one source and almost exclusively meteorological data, which were clearly written in that source, allowing for its flawless digitisation. We also disagree that the article's objectives are not defined – the reviewer even quoted them at the beginning of his review.

We use standard meteorological data analysis methods commonly used in climatology and meteorology. We are unsure which historical methods the reviewer is referring to. Collaboration with historians and the use of historical analysis methods are crucial in studies using verbal descriptions of weather conditions. We did not use verbal descriptions in this work; however, the article by Borm et al. (2021) is an excellent example of this type of work.

Borm, J., Kodzik, J., and Charbit, S.: Producing and communicating natural history in the long 18th century: Moravian observations concerning Greenland's climate in unpublished sources, Polar Record, 57, 1–10, https://doi.org/10.1017/S0032247421000218 , 2021.

The state of the art contains factual inaccuracies and is incomplete. For example:

The archives consulted are vaguely referred to as "many European and Canadian archives and libraries," without precise identification.

Reply: This passage of the text concerns research in archives and libraries used to write the article by Przybylak et al. (2010) and has no connection with this article.

Przybylak, R., Vízi, Z., and Wyszyński, P.: Air temperature changes in the Arctic from 1801 to 1920, Int. J. Climatol., 30, 791–812, https://doi.org/10.1002/joc.1918, 2010.

The authors mention "measurements for the expedition year September 1767–July 1768," although no expedition took place during that time. Information about who collected the data, for what purpose, and in which context is missing—despite the fact that these aspects are well-known in existing research.

Reply: Thank you for your suggestion. We changed the word "expedition" to "period".

We introduced the term "expedition year" in our articles, including this one, to properly analyse as much data as possible. Please note that, prior to the launch of regular measurements in the Arctic in the second half of the 19th century, meteorological observations were conducted during organised exploratory expeditions and later also during scientific expeditions. To reach the research area in the Arctic, expeditions usually started in the summer and finished their activities in the same year (or in the following years), also in the summer, when ice conditions allowed ships to sail. In such a case, to obtain an annual series, it must be started not on January 1 (calendar year) but immediately after the start of the research (meteorological observations in our case), i.e. most often in August or September. Therefore, we introduced the term "expedition year".

We would like to point out that meteorological observations from this year (i.e., September 1767–July 1768) are not the subject of analysis in this article. We mentioned them as part of the summary of the state of knowledge regarding available meteorological data and their treatment for the study area in the *Introduction* section. On the other hand, information about this series can be found in numerous published articles: Demarée et al. 2020, Demarée and Ogilvie 2021, Przybylak et al. 2024.

Demarée, G. R., Ogilvie, A. E. J., and Mailier, P.: Early meteorological observations in Greenland and Labrador in the 18th century: a contribution of the Moravian Brethren in: Proceedings of the 35th International Symposium on the Okhotsk Sea and Polar Oceans (2020), Mombetsu-2020 Symposium, 16–19 February 2020, OSPORA – Okhotsk Sea and Polar Oceans Research Association, Mombetsu, Hokkaido, Japan, 35–38, 2020

Demarée, G. R. and Ogilvie, A. E. J.: Missionary Activity in Greenland and in Labrador: The Context for Early Meteorological Observations, in: Legacies of David Cranz's 'Historie von Grönland' (1765), Christianities in the Trans-Atlantic World, edited by: Jensz, F, and Petterson, C., Springer, 141–164, https://doi.org/10.1007/978-3-030-63998-3_7, 2021.

Przybylak, R., Singh, G., Wyszyński, P., Araźny, A., and Chmist, K.: Air temperature changes in SW Greenland in the second half of the 18th century, Clim. Past, 20, 1451–1470,  https://doi.org/10.5194/cp-20-1451-2024 , 2024

There are also several factual errors and speculative statements regarding the Moravian Brethren. For instance, the authors write:

"They are usually available as meteorological registers (some have survived and are available in archives in Germany, the UK and the USA; see Demarée and Ogilvie (2008) for more details), and some have also been published in annual reports (diaries handwritten in Old German or English)."

Reply: We do not see a basis for some of the reviewer's statements. Our reply is below, point by point.

In reality:

   Only one set of measurements from Neu-Herrnhut is known for the second half of the 18th century.

Reply: Not true. Please find a quote from our earlier paper, published in 2024 (Przybylak et al. 2024), which analysed data for Nuuk (Godthaab) for the late 18th century:

'The thermal conditions of south-western Greenland in the second half of the 18th century were estimated using two unique series of meteorological observations. The first series (Neu-Herrnhut, 1 September 1767 to 22 July 1768, hereinafter 1767–1768) is the oldest long-term series of instrumental measurements of air temperature available for Greenland. The second (Godthaab, September 1784 to June 1792) contains the most significant and reliable data for Greenland for the study period'. See also Fig. 1 in this paper.

In addition, please also see one excerpt from the work of Demarée and Ogilvie (2021)

The Danish missionary Andreas Ginge (1754–1812) carried out more detailed meteorological observations at Godthåb (previously part of the Danish Royal Mission, now Nuuk) from September 1784 until July 1785 for minimum and maximum monthly values of the atmospheric pressure and air temperature, as well as three-times-a- day observations from October 1786 until June 1787.15 However, Brasen's meteorological observations (series 1767-68, authors suplement) precede Ginge's observations by approximately 20 years and thus constitute the first long-term meteorological time-series for Greenland.'

   The archival materials are not listed precisely.

Reply: Thank you for this suggestion. We have added more information about the source, including a link to the Royal Society archive webpage that provides a description of this source. See below these sentences added to Section 2.

'Currently, they are available in the manuscript MA/154, located in the archives of the Royal Society in London. ….. The title of the manuscript is: Meteorological observations at Godthåb [Nuuk], Greenland, by Charles Lewis Giesecke. More details about this source are available in https://catalogues.royalsociety.org/CalmView/Record.aspx?src=CalmView.Catalog&id=MA%2f154&pos=1.'

   The Moravian missionaries did not produce "annual reports" before the mid-19th century, and this varied by region and mission station.

Reply: Not true. We are attaching some of them to this text. Although these documents were not formally titled 'annual reports', they served as de facto official reports on the activities of the Moravian Brethren in a given settlement community. The attached documents provide reports of the Moravian Brethren for Lichtenau and Neuherrnhut, covering the years 1815–24 and 1796–1820, respectively, month by month. For clarity of the text, we, however, decided to change the text to:

…and some have also been included in texts of reports usually prepared for each month of the year and most often sent annually by ship to Europe (diaries handwritten in English or in Old German).

[Figure]

[Figure]

[Figure]

Diarium

der Gemeine

zu Neuherrnhut

1t August

1796 — 1820

B. 15. J. b. J. Midae

[Figure]

[Figure]

The term "Moravian handwritten diaries" is imprecise and does not clarify what kind of documents are meant.

Reply: According to us, the information is precise, see the example attached to the previous claim. The titles of these documents start with the word "Diarium …", as in the attached example. We have made a minor adjustment to the text in response to the previous claim, so the text should now be clearer for the Reviewer.

It remains unclear what the authors mean by "meteorological registers," since the Moravians did not use such terminology.

Reply: Of course, they did not use such terminology, but such a description of meteorological data ordered in tables is often used in the English literature, so we use this terminology. But if this is somehow, according to the reviewer, not appropriate, we propose to change this to:

…and some have also been included in texts of reports usually prepared for each month of the year and most often sent annually by ship to Europe (diaries handwritten in English or in Old German).

The references to "archives in the UK and USA" are vague; if the authors mean Moravian archives, these mainly contain copies of original documents preserved in Germany.

Reply: For clarity, according to the Reviewer's suggestion, we have added more precise information; now the text is:

They are usually available in the form of tables with meteorological data. Some have survived and are available in archives in Germany (The Unity Archives – Moravian Archives Herrnhut), the UK (The Library and Archives of the Royal Society and Moravian Church Archive and Library in London) and the USA (Moravian Archives Bethlehem).

The use of the term "Moravian Brothers" is incorrect.

Reply: Thank you very much for this remark. We changed to Moravian Brethren.

The Moravians generally did not collect data for scientific purposes. Their possible involvement in Giesecke's measurements could, however, be verified in the relevant archives. Before Giesecke's arrival, Moravian meteorological observations in Greenland were limited to their own use—this is well documented in previous research.

Reply: The statement in the first sentence of the review is generally true. However, we also have information that they performed meteorological observations on behalf of scientific institutions in Labrador (Deutsche Wetterdienst) or scientific societies (Palatine Meteorological Society in Mannheim) in Godthaab. See the passage of text from Przybylak et al. 2024:

*'The second series of measurements, although not continuous (see Fig. 2), is the greatest and most reliable one available for Greenland for the study period. Observations were made three times a day (07:00, 14:00 and 21:00 LT) by the Danish Reverend Andreas Ginges (1754–1812) using a methodology and instruments provided by the Palatine Meteorological Society. The sub-daily or daily air temperature data exist for the following periods: September 1784 to June 1785, January–June 1787, November–December 1788 and January 1790 to June 1792 and are available in the manuscript entitled "Astronomiske og meteorologisk Iagttagelser, anstillede i Godthaab i Grønland 1782–1792" and in the society's yearbook Ephemerides Societatis Meteorologicae Palatinae'*

No specific old English script exists, in contrast to the old German script

Reply: You are right, but we did not write such a sentence in our manuscript. Nevertheless, for clarity, we changed this sentence to: ….*(diaries handwritten in English or in Old German*

The claim that the series from September 1767 to July 1768 represents the "oldest meteorological data for Nuuk" is also inaccurate, as the location was Neu-Herrnhut (now within modern Nuuk but not historically identical). Furthermore, the dataset analyzed is not the longest pre-1840 meteorological series for the Arctic—the authors do not clarify which definition of "Arctic" they apply.

Reply: In our opinion, because Neu-Herrnhut lies within the current boundaries of Nuuk, our statement is correct. When meteorological stations are located within a given settlement or city, they change location due to various factors (e.g., loss of representativeness); therefore, the reviewer's approach of treating individual observation series separately is unacceptable. Please see also the statement written by Demarée et al. (2021):

*'Christopher Brasen (1738 1774) carried out meteorological observations at Neu Herrnhut [modern day Nuuk] from 1 September 1767 through 31 July 1768. As such, these observations constitute **the oldest long-term meteorological observational time series for Greenland.'***

The following sentence was added to clarify the definition of the Arctic:

*The Arctic is defined as the region after* Atlas Arktiki *(Treshnikov 1985, see also Fig. 1.1 in Przybylak 2016).*

For such an Arctic region, our statement is true; of course, not for definitions that describe the Arctic as an area lying to the north of 60°N, 65°N, and so on.

Demarée, G. R. and Ogilvie, A. E. J.: Missionary Activity in Greenland and in Labrador: The Context for Early Meteorological Observations, in: Legacies of David Cranz's 'Historie von Grönland' (1765), Christianities in the Trans-Atlantic World, edited by: Jensz, F, and Petterson, C., Springer, 141–164, https://doi.org/10.1007/978-3-030-63998-3_7, 2021.

The manuscript is not, in fact, a newly discovered source. It was already known and received by scholars in the 19th century and remains known in current research (see Schmidt 2020).

Reply: The reviewer is correct; in the 19th century, this manuscript was likely known, but we did not find any information about its use at that time. Even in a recent publication (Vinther et al., 2006), which used all early instrumental series for reconstructing air temperature in SW Greenland, this series was not included, meaning it was unknown to the authors. Also, in quite a new publication published by the Danish Meteorological Institute (J. Cappelen ed, 2021, DMI Report 21-04, https://www.dmi.dk/fileadmin/Rapporter/2021/DMIRep21-04.pdf) entitled: *Greenland – DMI Historical Climate Data Collection 1784-2020,* this series is not mentioned, and there is gap shown for years analysed in our manuscript in the data, **see Fig. 22 (page 50) in this report**.

Unfortunately, we are unable to locate the paper mentioned by the Reviewer. Please provide the exact reference information for the publication. Providing only the surname, which is a common German surname, prevented us from locating this publication. Also, John Cappelen from DMI (the best specialist in knowledge about available meteorological data for Greenland) did not find this publication.

Not all references cited in the text appear in the bibliography, and the archival sources themselves are not listed there.

Reply: Thank you for this information. We corrected all the missing information.

The historical source is not described in detail; even the original title is omitted. The authors rely on a copy available in London without referencing or verifying it against the original manuscript from Denmark and other existing copies. They also appear unaware that Giesecke produced two manuscripts containing meteorological observations from  his Greenland travels. No comparison is made with Giesecke's travel diary—published several times in the 19th and 20th centuries—which contains relevant meteorological data and the location of the thermometer.

Reply: In response to earlier Reviewer's comments, we provided additional information about the source of the use. In our case, since all the meteorological data were easily readable and the copy was complete, we decided there was no need to verify it against the original. This is especially true since the originals need to be protected as much as possible to prevent damage.

Shorter observation series from various places where Giesecke travelled, not collected from permanent observation points, are not useful for our purposes in the present manuscript. However, they are important and may be useful for future analyses using scattered observation series from different locations in SW Greenland.

The German text at the bottom of the table is neither transcribed nor analyzed.

Reply: Not true, the important fragment for the DDTV analysis is given in the original (translated by a historian) and translated into English. The text is attached below:

*Under the table of data from February 1812, Giesecke wrote the following (see Fig. 3): "Hochste Kälte 21sten - 23sten, 24sten und 25sten [Februar] 36 ½ °. Wärmer den 26ten – 1 ½ °" [translation, first author: Extremely cold 21st – 23rd, 24th and 25th [February] -36.5 °C. Warmer the 26th – 1.5 °C.].*

In summary, the paper lacks a solid historical and archival foundation. Due to historical methodological weaknesses and factual inaccuracies, the object of analysis is not clearly defined, the author of the manuscript not identified, and the reliability of the scientific results is uncertain. A thorough methodological revision, including proper archival research and accurate use of historical sources, is strongly recommended to establish a solid basis for scientific analysis.

Reply: This is not a historical work, but strictly a climatological one. We are climatologists, and in this article, we only analyse meteorological measurements **contained in a single source**, which provides all data that is clear and complete.

The review contains no substantive objections to the analyses of the meteorological data used. All objections presented concern the Introduction section (literature overview) and the description of the source. Thank you very much for these comments, as they also helped us significantly improve the article's content.

---

## Author Comment (AC4)

**Reply to Reviewer 1** (in green font)

Anonymous Referee #1, 20 Oct 2025
Summary: The manuscript presents an analysis of a newly discovered series of historical meteorological data comprising subdaily air temperature, wind and pressure taken by Charles Lewis Giesecke in Nuuk, Eastern Greenland, covering the period 1809-1813. The authors compare these observations with recent observations from the same location, analyse long-term changes in temperature and its connection to wind direction.

The main findings are that air temperature was almost always colder than in recent times, and that advection from the North-East played a more significant role than it does today. The colder temperatures agree with other indirect information derived from ice-core records.

Recommendation:

The study is interesting, as meteorological observations this old are very rare. The manuscript is, in my opinion, well and clearly written, although a few figures could be more clearly designed. There are some aspects that the study does not cover, and that could also be interesting for the reader, as I explained in more detail below. My recommendation is that the manuscript can gain from some moderate revisions, which are certainly feasible

Reply: Thank you very much for your suggestions. All were taken into account.

Main points:

1) Clarity of some figures. Figure 5, perhaps one of the most relevant in the study, is not optimally designed, making it difficult for the reader to skim the relevant information. My suggestion is to display the 1D and 2D spreads of the modern temperatures as coloured surfaces in the background, against which the mean of the modern and of the historical temperatures is plotted as dark lines
Reply: Done

2) Similarly, Figure 9 could include the temperature levels as circles, instead of a linear scale on the y-axis as it is now.
Reply: Done

3) Perhaps more importantly, the study is strongly focused on the mean annual cycle, and essentially all figures display in some way or another the mean annual cycle derived from the 5 years of observations. No figures actually show a time series over the period of observations, such as monthly or annual means. The period is admittedly short, but this type of information would be useful when discussing the purported impact of volcanic eruptions. For instance, this period includes the 1809 Tambora (?) eruption, less known than its 1815 counterpart, but nevertheless intense. The time series of annual means or monthly anomalies might provide insights into the impact of this eruption on temperature in Nuuk and its recovery in the following years. Also, the lack of any clear signal would be relevant. A time series of wind direction frequency could also be interesting, as eruptions have been suggested to impact the state of the NAO towards a more zonal state. Would this impact be visible in the wind direction data? Again, a positive or negative answer would be, in my view, interesting.

Reply: Thank you for this suggestion. Two new figures have been added (Figs. 5 and 6). The following text was added to the Result part:

*Figure 5 shows the course of monthly average air temperatures in the study period to determine whether the strong eruption (the third largest since 1500) of an unidentified volcano at the end of 1808, and most likely at the beginning of 1809 (Vinther et al. 2006; Timmreck et al., 2021), influenced the air temperature in SW Greenland. A significant cooling in Nuuk occurred from the second winter after the eruption. This cooling occurred mainly in wintertime and persisted until the end of the series, but the greatest was in the expedition year 1810/11. Following the eruption, there was a notable increase in the frequency of calms and winds from the north-east sector, accompanied by a decrease in the number of winds from other directions, particularly those from the south-east and south-west sectors (Fig. 6).*

And to the Discussion and conclusions part:

*Our results presented in Fig. 5 confirm the existing finding of Vinther et al. (2006), i.e. the very cold weather in the first years of the 1810s. According to our data, an observed cooling occurred after 1809, initially small in winter 1809/10 and then significantly greater in the following three winters, 1810/11-1812/13. This cooling is also observed in data from paleoreanalysis (ModE-RA), but not in the data from the 20th Century Reanalysis (20CRv3). In summer, some cooling is evident only in 1811. As a result, the coldest expedition year after a volcanic eruption in 1809 occurred in 1810/11. Data series presented in Fig. 5 (except 20CRv3) also confirm Schneider et al.'s (2017) finding that after the 1809 event, temperatures remain low until the Mt. Tambora eruption in 1815. Changes in atmospheric circulation that occurred after this eruption, from a negative to a positive NAO indices in wintertime (see Luterbacher et al., 2002a, b), caused a change in wind directions in SW Greenland, i.e. (domination of winds from the northern sector, see Fig. 6). Positive phase of the NAO cause a significant decrease (3-6°C) of winter air temperature in the western part of Greenland in contemporary series of data, particularly in its SW part (see Fig. 12 in Przybylak 2000). A similar finding was recently presented by Faust et al. (2025, submitted; see their Fig. 4) for southwest Greenland, based on temperature reconstruction using a high-resolution sedimentary record, for the late Holocene.*

[Figure]

**Fig. 5 (a)** Time series of mean monthly air temperatures in Nuuk in the period 1807–13 according to different datasets, and **(b)** their anomalies in reference to the period 1991-2020. A large, unknown, volcanic eruption most likely occurred in early 1809.

[Figure]

**Fig. 6.** Frequency of winds in Nuuk in the winter season (Dec – Mar) in particular years of the period 1807–13. In black, the year of the eruption of an unknown volcano

4) Following this time-series approach, another suggestion is to compare the monthly anomalies or annual means with those from the neighbouring cell in the 20CR reanalysis. The agreement probably cannot be expected to be good, but it would also be an interesting test for the 20CR reanalysis using independent historical observations.
Reply: Done, see new Fig. 5.

Particular points:

4) 'A cooling of this severity has previously been found for the study region, the whole of Greenland and the whole Arctic.'

The meaning of this sentence in the abstract is not clear to me, unless it refers to previous studies (?). If yes, please state it so.

Reply: Done

A cooling of this severity in the first decades of the 19th century for the study region, the whole of Greenland and the whole Arctic has also been earlier reconstructed by other scientists using different proxy data and models.

5) 'Intense volcanic activity and, to a lesser degree, the low solar activity connected with the Dalton minimum are most often given as reasons for the cooling of the early 19th century.'

I would be reluctant to include this sentence in the abstract, as it is actually not a conclusion of the present study. It can mislead the reader into thinking that this study also attributed the cooling to those climate forcings.

Reply: Thank you for this suggestion. You are right, this is based on the literature review. The sentence was deleted from the Abstract.

6) 'about the existence of long-term continuous meteorological observations'

What does 'continuous' mean here ? I guess daily temporal scales, but it is unclear.

Reply: Done. The sentence was changed to: 'about the existence of a long-term continuous series of daily and sub-daily meteorological observations'. We hope that the sentence is clearer now.

7) 'We calculated MDATs according to eight different formulas:'

I would have a suggestion that the authors may want to follow, although it is not critical. Instead of testing different links between subdaily measurements and daily means, a linear regression would yield the proper weights for the three subdaily data to reconstruct the daily mean

Reply: We can't really use linear regression because we don't know the exact times of measurement in the historical series. Therefore, we tested various possible variants of measurement times, including the morning, noon, and evening parts of the day to estimate the biases in comparison to the so-called real daily mean temperature (calculated from 24 measurements a day). If we knew the times of observation, we could perform a linear regression between the 24-hour average and the average of the known three measurement hours based on contemporary data and apply the regression formula to the historical series.

8) 'temperature between 5 and 25 February never rose above -20 °C, and between 19 and 25

February it was even above -30 °C'

I guess the authors mean that the temperature remains above -30 °C. The formulation is ambiguous

Reply: Yes. You are right. For clarity, we propose the following change to the sentence:

'temperature between 5 and 25 February never rose above -20 °C, and between 19 and 25 February it was constantly below -30 °C'

9) 'The question arises: What could be the reason for such great heating from one day to

the next? '

The authors suggest that föhn was responsible for this sudden warming. It seems plausible, but perhaps

One analogous situation can be found in 20th-century observations to support this hypothesis, possibly dating back to before the retreat of the ice sheet.

Thank you for this suggestion. We have conducted a search for such cases and concluded that.

although the entire series of temperatures from Nuuk (1873–2023) reveals that the maximum MDAT change from one day to the next was of 20.7 °C (from 7 to 8 March 1939), we cannot assume that the historical value of 35.6 °C is incorrect.

10) line 410 , consider opening subsections for wind, daily temperature variability, etc. It would later help the reader skimming the article.

Reply: Done, see the text

11) 'In historical times, the irregularity of thermal roses is..'

Perhaps not irregularity, but the deviation from circular symmetry

Reply: Thank you for the suggestion. We have changed 'irregularity' to your proposition, i.e., *deviation from circular symmetry*.

- **Citation**: https://doi.org/10.5194/egusphere-2025-4313-RC1
- **RC2**: 'Small correction on my previous comment', Anonymous Referee #1, 20 Oct 2025 reply

I meant, of course, 'Nuuk, Western Greenland'

Reply: Thank you.